# Trajectories of Mental Distress and Resilience During the COVID-19 Pandemic Among Healthcare Workers

**DOI:** 10.3390/healthcare13050574

**Published:** 2025-03-06

**Authors:** Andreas M. Baranowski, Simone C. Tüttenberg, Anna C. Culmann, Julia-K. Matthias, Katja Maus, Rebecca Blank, Yesim Erim, Eva Morawa, Petra Beschoner, Lucia Jerg-Bretzke, Christian Albus, Kerstin Weidner, Lukas Radbruch, Cornelia Richter, Franziska Geiser

**Affiliations:** 1Klinik und Poliklinik für Psychosomatische Medizin und Psychotherapie, Universitätsklinikum Bonn, Universität Bonn, 53127 Bonn, Germany; 2Klinik für Palliativmedizin, Universitätsklinikum Bonn, Universität Bonn, 53127 Bonn, Germany; 3Systematische Theologie, Evangelisch-Theologischen Fakultät, Universität Bonn, 53121 Bonn, Germany; 4Psychosomatische und Psychotherapeutische Abteilung, Universitätsklinikum Erlangen, 91054 Erlangen, Germany; 5Klinik für Psychosomatische Medizin und Psychotherapie, Klinikum Christophsbad, 73035 Göppingen, Germany; 6Klinik für Psychosomatische Medizin und Psychotherapie, Universitätsklinikum Ulm, 89081 Ulm, Germany; 7Klinik und Poliklinik für Psychosomatik und Psychotherapie, Universitätsklinikum Köln, 50937 Köln, Germany; 8Klinik und Poliklinik für Psychotherapie und Psychosomatik, Universitätsklinikum Dresden, 01307 Dresden, Germany

**Keywords:** resilience trajectories, healthcare workers, COVID-19 pandemic, longitudinal study, growth mixture modeling (GMM), stress responses

## Abstract

**Background/Objectives**: The recent COVID-19 pandemic posed a significant psychological challenge for healthcare workers. Resilience and the extent of psychological stress varied across professional groups and individual circumstances. This study aims to longitudinally capture the trajectories of psychological stress and resilience among medical personnel during the pandemic and identify various contributing factors. **Methods**: Over a period of three years, healthcare workers from five locations (Bonn, Cologne, Ulm, Erlangen, and Dresden) were surveyed regarding their psychological stress (PHQ-4) and other aspects of mental health. Data were collected at five different points during the pandemic. Using Growth Mixture Modeling (GMM), various stress trajectories during the crisis were modeled without initial adjustment for covariates to allow for an unbiased identification of latent classes. Differences in demographic and occupational factors (e.g., age, gender, profession) were analyzed across the identified trajectory groups in subsequent steps. **Results**: The application of GMM revealed three distinct profiles of psychological stress and resilience among the respondents, largely consistent with the literature. The largest group was the ‘resilience’ group (81%), followed by the ‘recovery’ (10%) and ‘delayed’ groups (9%). Group membership was consistent with self-reported trajectories over the course of the pandemic. It was not possible to predict individual trajectories based on the results of a short resilience questionnaire (RS-5). **Conclusions**: The COVID-19 pandemic had multiple psychological impacts on healthcare workers, manifesting in clearly differentiated group trajectories of distress over time. While a majority of respondents in this sample exhibited a stable trajectory with low distress, other groups showed varying stress responses over time. These findings highlight the necessity of longitudinal approaches to understand the complex interplay of stressors and coping mechanisms during prolonged crises.

## 1. Introduction

The COVID-19 pandemic posed a profound threat to global wellbeing, extending beyond the direct effects of infection with the SARS-CoV-2 virus. Public health measures, such as lockdowns, social distancing, and changes in work and school routines, significantly disrupted daily life [1,2]. Early in the pandemic, the potential for a substantial impact on mental health was recognized, and subsequent studies have confirmed elevated levels of distress, depression, and anxiety compared to pre-pandemic times [3,4,5]. However, the impact on mental health was not uniform, with certain populations experiencing more severe effects [6].

Among the groups most severely impacted by the pandemic are healthcare workers [7,8]. These individuals faced extraordinary pressures, including a high risk of exposure to the virus, increased workloads, and the emotional toll of caring for severely ill patients. This cumulative stress made healthcare workers particularly vulnerable to mental health issues [9,10]. Despite their critical role in managing the pandemic, there is a need for more focused research on their psychological wellbeing and the factors that can help mitigate adverse outcomes.

Resilience is a crucial concept when examining the psychological health of healthcare workers during the COVID-19 pandemic. Resilience can buffer the negative effects of stress and contribute to better mental health outcomes [11]. Understanding how resilience operates in this high-stress environment can provide valuable insights into support mechanisms that can enhance the wellbeing of healthcare workers [12].

While numerous studies have provided snapshots of mental health status during the COVID-19 pandemic, insights from a single point in time are inherently limited [13]. Mental health is dynamic, with individual experiences evolving in response to ongoing challenges and changing circumstances [14]. Longitudinal studies are essential for capturing these individual trajectories over time. Such studies can identify patterns of resilience and vulnerability, offering a more comprehensive understanding of mental health dynamics and informing more effective support strategies. This is particularly crucial for populations highly exposed to chronic stress and trauma, such as healthcare workers, who may experience long-term psychological consequences that extend well beyond the initial crisis.

Galatzer-Levy et al. [15] described distinct trajectories based on a meta-analysis of 54 studies on individual-level macro-stressors (i.e., job loss, major health event) in the general population. They found the most frequently reported trajectories were ‘resilience’, ‘chronic’, ‘recovery’, and ‘delayed-onset’ trajectories, with a high variance depending on the initial event and the population studied. The resilience trajectory is characterized by individuals who maintain stable, low levels of psychological distress despite exposure to significant stressors. The ‘recovery’ trajectory includes those who initially exhibit high levels of distress but gradually return to baseline functioning over time. The ‘chronic’ trajectory is marked by sustained high levels of distress without significant improvement. The ‘delayed-onset’ trajectory consists of individuals initially coping well but experiencing increased distress later. Building on this framework, Schäfer et al. [16] adapted these trajectories to the context of the global COVID-19 pandemic, identifying similar patterns of resilience and distress among various populations as for individual-level macro-stressors. Both overviews emphasize the high heterogeneity of the trajectories between studies.

Our study aims to explore the resilience trajectories of healthcare workers over the course of the COVID-19 pandemic. By conducting a longitudinal analysis with data collected at multiple time points, we seek to identify distinct patterns of resilience and their relationship with symptoms of anxiety and depression. Our goal is to provide evidence that can inform targeted interventions to support the mental health and resilience of healthcare workers, ultimately enhancing their ability to cope with ongoing and future public health crises.

## 2. Materials and Methods

### 2.1. Data Collection

The study was conducted by the psychosomatic departments of university hospitals in Bonn, Erlangen, Ulm, Cologne, and Dresden, Germany. The participation link was provided through online platforms and mailing lists of the participating university hospitals as well as other general hospitals and several medical professional associations. The entire questionnaire took 20 min to complete and was approved by the ethics committees of all participating medical faculties. All participants provided their online informed consent prior to completing the survey.

Adjustments were made to the items to reflect the ongoing pandemic situation at each data collection point. The complete questionnaire, conducted entirely in German, covered various topics including demographic details such as age and gender, as well as professional data including occupation type, years of working experience, and employment status, among others. Our analysis primarily focused on age, gender, profession, and the questionnaires described in the presented measures. Unipark (www.unipark.com) and SoSci Survey (www.soscisurvey.de) were used to program and host the survey. The inclusion criteria for participation were a minimum age of 18 years, working in the healthcare sector, residing or working in Germany, and having proficient German language skills. Meanwhile, the exclusion criteria were being under the age of 18, not working or no longer working in the healthcare sector (e.g., due to retirement or leaving the profession), residing outside of Germany without professional ties to the German healthcare system, or lacking sufficient German language proficiency to complete the survey.

The online survey was conducted between 2020 and 2023 and included five measurements. The measurement time points were 20 April 2020–5 July 2020 (T1), 17 November 2020–7 January 2021 (T2), 28 May 2021–16 July 2021 (T3), 7 February 2022–1 May 2022 (T4), and 17 April 2023–4 June 2023 (T5). The first survey was conducted during the first wave of the pandemic. The following three measurement time points corresponded approximately to the second, third, and fourth waves of COVID-19 infection numbers in Germany [17]. The fifth and last data collection occurred at the end of the fifth COVID-19 wave, after two years of the pandemic, with many public safety measures starting to loosen up in Germany.

### 2.2. Sample Characteristics

The study included participants who took part in at least three of the five data collection points. This resulted in 2973 individual observations and 910 participants. Of the 910 participants included in the final analysis, 543 participated at T1, 676 at T2, 640 at T3, 704 at T4, and 506 at T5. Due to staggered enrollment, where participants could enter the study at T1, T2, or T3 and were required to complete at least two subsequent assessments, participant numbers at each time point reflect both continued participation and new enrollments rather than a fixed cohort with traditional attrition. The majority of the sample consisted of women (*n* = 701), with 224 men and 5 people who identified as diverse. The participants who identified themselves as diverse were included in all analyses except for those looking at gender differences, because the sample size was too small for a meaningful analysis. Age was assessed based on 5 groups, with the majority falling in the age range of 51–60 (*n* = 325), followed by ages 31–40 (*n* = 268), 41–50 (*n* = 256), 18–30 (*n* = 170), and >60 (*n* = 69). Participants were placed in six occupational groups, based on their self-disclosure: physicians (*n* = 168), nurses (*n* = 231), medical technical assistants (*n* = 156), psychologists (*n* = 50), pastoral care workers (*n* = 58), and others (*n* = 407). Others consisted of a wide range of professions, including students, administrative staff, physiotherapists, and social workers, and served as a general reference group (see Table 1).

### 2.3. Measures

#### 2.3.1. Depressive and Anxiety Symptoms (PHQ-4)

Symptoms of depression and generalized anxiety during the prior two weeks were evaluated using the Patient Health Questionnaire PHQ-4 [18]. This instrument includes four items, including “Over the last 2 weeks, how often have you been bothered by feeling nervous, anxious or on edge?” and “Over the last 2 weeks, how often have you been bothered by feeling down, depressed, or hopeless?” Responses were recorded on a Likert scale that ranged from 0 (“not at all”) to 3 (“almost every day”). In our analyses, these outcomes were treated continuously, but scores above a cut-off of ≥4 are optimal for detecting any depressive or anxiety disorder and ≥6 to detect major depressive and generalized anxiety disorder [19]. Cronbach’s α in this sample over the five measure points ranged from 0.84 to 0.86.

#### 2.3.2. Resilience (RS-5)

We utilized the five-item Resilience Scale (RS-5) [20] to measure resilience, which is a condensed version of the original 25-item scale by Wagnild and Young [21]. Resilience is conceptualized by the authors as a positive personality trait that promotes individual adaptability and is encapsulated within two primary dimensions: acceptance of self and life and personal competence. An example of an item from the scale is “Keeping interested in things is important to me”. Participants rated their agreement with each statement using a seven-point Likert scale, ranging from 1 (‘No, I disagree’) to 7 (‘Yes, I completely agree’). In our study, Cronbach’s α ranged from 0.78 to 0.83 during the five measure points.

#### 2.3.3. Self-Reported Trajectories of Resilience

At T5, we included drawings of six curves of stress trajectories throughout the pandemic in the questionnaire and asked participants to choose which of these represented best their experience. Each drawing had a description; 1. ‘The pandemic initially caused me stress, but over time my stress level normalized again and returned to its original level’, 2. ‘The pandemic initially caused me little or no stress, and my stress level quickly returned to its original level’, 3. ‘The pandemic initially caused me stress, but I was able to grow as a result and feel stronger and more resilient today than before the outbreak of the pandemic’, 4. ‘The pandemic initially caused me stress, it got better over time but my stress level is still higher than before the outbreak of the pandemic‘, 5. ‘The pandemic initially caused me stress and I have remained at this stress level to this day’, and 6. ‘The pandemic initially caused me stress, which became more and more severe over time, so that I am currently suffering from very high levels of stress’. Participants were also able to check ‘None of the trajectories apply to me’. Only participants who answered the questionnaire at T5 could be included in this analysis, which amounted to about 20% (*n* = 183) of the total sample.

### 2.4. Statistical Analysis

All statistical analyses were conducted using IBM SPSS Statistics (Version 26) and R (Version 4.4.1). The internal consistency of the scales used and described above was measured with Cronbach’s α. For descriptive and comparative statistics, the chi-square test of independence and analyses of variance (ANOVA) were performed and the effect size was given in partial η^2^. In cases of multiple comparisons, Tukey post hoc tests were conducted and the effect size was given in Cohen’s d. The significance level for all statistical tests was set at *p* < 0.05.

We utilized growth mixture models (GMMs) to discern distinct subgroups of participants who showed varying trajectories in their anxiety and depression symptoms. GMMs build upon latent growth curve models (LGCMs) and are implemented within a structural equation modeling context [22]. LGCMs enable the modeling of repeated measures of a directly observed variable (e.g., symptoms of anxiety or depression) through the use of latent variables that capture the intercept (initial level of the observed variable) and the slope (change across time). GMMs further expand on this approach by facilitating the identification of subgroups (‘latent classes’) that display unique intercepts and slopes, thus showcasing different patterns in symptom trajectories over time.

LGMM analyses were conducted in R with the lcmm package for LGMMs (random intercepts and slopes). We tested one to six trajectories, selecting the optimal model based on the Akaike information criterion (AIC), the sample-size-adjusted Bayesian information criterion (BIC) [23], the Likelihood Ratio Test (LRT), and the Vuong–Lo–Mendell–Rubin (VLMR) test [24,25]. For AIC and BIC, lower values indicated a better fit. Models demonstrating an improved fit with an additional trajectory that included a low proportion of participants (<5%) were rejected in favor of a more parsimonious solution [26].

## 3. Results

### 3.1. Trajectories of Resilience

We first tested a two-class model, which resulted in AIC = 5303.12 and BIC = 5356.07. The three-class model (AIC = 5024.40, BIC = 5096.60) demonstrated a substantial improvement in fit. The LRT statistic comparing the two-class and three-class models was 286.72 (*p* < 0.001), and the VLMR test statistic was also 286.72 (*p* < 0.001), both strongly favoring the three-class solution. As additional models beyond three trajectories resulted in small class sizes (<5% of participants), we selected the three-class model as the most parsimonious and best-fitting solution.

Given this statistical justification, we proceeded with the interpretation of the three identified resilience trajectories. These trajectories have been described before [19,20] and in this study were named in accordance with the literature (‘resilience’, ‘recovery’, and ‘delayed’). Figure 1 presents the three-class trajectories of resilience, measured by symptoms burden of anxiety and depression. The three classes can be characterized as follows:

The ‘resilience‘ trajectory (*n* = 740, ≙ 81%) included the majority of respondents who reported low symptoms throughout the pandemic, remaining below clinically relevant levels of anxiety and depression. Mean values of symptoms for this group increased slightly towards the summer of 2021 (T3), with M = 2.64, SD = 0.45, but declined again thereafter.

The ‘recovery’ trajectory (*n* = 87, ≙ 10%) was characterized by respondents who exhibited the highest symptom levels at the second measuring point in 2020 (T2), with M = 8.28, SD = 0.55 (cut-off ≥ 6 for major depressive and generalized anxiety disorders). Following this peak, there was a consistent decline in symptom levels across subsequent measurements. By the final measurement point in 2023 (T5), participants on average only showed mild symptoms below the cut-off for detecting any depressive or anxiety disorders (M = 3.96, SD = 0.60).

The ‘delayed’ trajectory (*n* = 83, ≙ 9%) initially exhibited mild symptoms of anxiety and depression, with M = 4.04, SD = 0.69. Over the course of the pandemic, these symptoms progressively increased, culminating in a peak during the spring of 2022 (T4), M = 8.64, SD = 0.63. These elevated symptom levels persisted even after a decline in COVID-19 cases in Germany and the relaxation of protective measures in the spring of 2023.

In order to increase the size of our database, we tentatively conducted the same analysis in a sample including participants who attended at least two (instead of three) time points (*n* = 3093). Here, in the place of one ‘recovery’ group, we identified two similar ‘recovery’ types of trajectories: one with higher and one with lower levels of distress severity. Additionally, there was a U-shaped trajectory, where participants exhibited a high level of symptom burden at the beginning of the pandemic (T1), which decreased over time to subclinical levels at T3, only to increase again towards the end (T5). However, this group comprised only 2% of the study population. Due to the constraints discussed in the methods section, these results, although clinically interesting, remain provisional until a replication can be performed. All further results reported stem from the sample with at least three longitudinal measurements.

### 3.2. Cohort Characteristics

Based on the identified trajectories, we further analyzed the groups. A chi-square test of independence was conducted to examine the distribution of gender, age, and profession across the various groups. We observed a significant difference in the distribution for age, X^2^ (10) = 19.29, *p* = 0.036, and profession, X^2^ (10) = 19.29, *p* < 0.001; however, the distribution for gender was not significant, X^2^ (4) = 8.38, *p* = 0.079. Given the significant outcomes of the chi-square tests for age and profession, standardized residuals were analyzed as a post hoc measure to identify specific categories contributing to the observed differences.

Post hoc analysis revealed that there are fewer physicians (*p* = 0.048) and more MTAs (*p* < 0.001) in the ‘recovery’ group than expected. Other professions did not show a significant difference in their distribution between the groups in the post hoc analysis. Although the chi-square tests indicated significant differences in the overall distribution among age groups, the post hoc analysis of residuals showed no significant discrepancies in age distribution between groups. This absence of significant post hoc findings for age suggests that the observed overall significance may be attributed to a cumulative effect of minor deviations across multiple categories, none of which individually reach the threshold for statistical significance.

### 3.3. Symptom Burden and Resilience

To determine whether baseline (T1) values in the short resilience questionnaire RS-5 could predict the trajectory groups, we conducted an ANOVA. The results indicated that the ‘resilience’ (M = 5.81, SD = 0.86) and ‘recovery’ (M = 5.15, SD = 1.13) groups differed significantly in their baseline RS-5 values (Tukey post hoc test, *p* = 0.037, ηp^2^ = 0.04), whereas no significant differences were found between the ‘delayed’ and the other groups. Furthermore, although the ‘recovery’ group initially exhibited lower resilience scores and higher symptoms, they eventually showed a significant reduction in symptoms, ending with a lower symptom burden compared to the ‘delayed’ group, where RS-5 scores were high initially, but symptoms increased over time. This pattern suggests that baseline resilience scores (RS-5) alone are not sufficient to predict future distress symptoms or symptom trajectories.

Given these findings, we further investigated the relationship between symptom burden and resilience/RS-5 scores at various points of measurement. Resilience is typically considered a stable trait, not significantly influenced by situational changes. However, our analysis revealed an instability of RS-5 scores depending on the groups, with a significant correlation between symptom burden (PHQ-4) and resilience scores (RS-5) in cross-sectional analyses at all time points. Specifically, as symptom burden increased, resilience scores tended to decrease, and vice versa (see Figure 2). This inverse relationship was observed in all three groups: ‘Resilience’ (*r* = −0.41, *p* < 0.001), ‘recovery’ (*r* = −0.40, *p* < 0.001), and ‘delayed’ (*r* = −0.26, *p* = 0.012).

### 3.4. Comparison of Self-Reported and Data-Based Stress Trajectories

In the next step, we analyzed the three groups in relation to the self-reported retrospective assignment to predefined stress trajectories, represented by six typical graphs (presented only at T5, see Figure 3). In the ‘resilience’ group (*n* = 155), 62% chose a graph depicting an initial phase of stress, after which stress levels returned to baseline or even lower (Figure 3A–C). However, about one-third of the participants in this group chose a graph with persistently elevated stress levels (Figure 3D–F). In the group with a delayed stress response (*n* = 14), 57% chose a graph depicting an elevated stress level at T5 (Figure 3D–F). Additionally, participants in the ‘delayed’ group were the least likely to find a matching predefined trajectory, with 21% selecting ‘None of the trajectories apply to me’. In the ‘recovery’ group (*n* = 14), 57% chose a graph depicting a return to a lower level after an initial phase of stress, while still remaining above the stress level of before the pandemic (Figure 3D) (see Figure 3).

The choice of graph was also associated with individual PHQ-4 scores at T5. Participants who chose a graph where stress quickly returned to baseline after initially low stress (Figure 3D) had the lowest symptom burden at T5 (*M* = 4.50, *SD* = 1.41). The group that chose a graph with high stress levels at the end (Figure 3E,F) also had the highest PHQ-4 scores at T5, with *M* = 10.20, *SD* = 4.55.

## 4. Discussion

In a longitudinal study with measurements conducted at five time points over three years during the COVID-19 pandemic, covering the five consecutive infection waves, we explored trajectories of mental distress among healthcare workers in Germany. This is the largest and longest running study of this kind to date. Symptoms of anxiety and depression showed various patterns, reflecting the dynamic nature of the pandemic and its impact on healthcare workers. Using growth mixture models, we identified three distinct trajectories of mental distress. The majority of participants were classified into the ‘resilience’ group, characterized by comparatively low symptoms of anxiety and depression throughout the pandemic. Another group, the ‘recovery’ trajectory, experienced high levels of distress initially, which gradually decreased over time, aligning with periods when the pandemic’s impact lessened. The ‘delayed’ trajectory showed a gradual increase in symptoms, peaking during the spring of 2022, indicating that some healthcare workers experienced heightened stress as the pandemic progressed.

When lowering the inclusion criteria from at least three to at least two longitudinal measurements, we identified an additional trajectory with a U-shaped pattern. Since we used Latent Growth Mixture Modeling with staggered enrollment, participants were not required to have data at three consecutive time points for this pattern to emerge. Instead, LGMM estimates latent trajectories using all available data points across individuals, allowing the model to detect underlying symptom progression even when participants contributed data at different time points.

This U-shaped trajectory was characterized by a high symptom burden at the beginning of the pandemic, a decline to subclinical levels by the third time point, and a subsequent increase toward the end of the study period. One possible explanation for this pattern could be that initial stress exposure led to temporary adaptation, but as the pandemic prolonged, accumulated stressors exceeded individuals’ coping resources, leading to renewed distress. This aligns with research suggesting that cumulative exposure to adversity can lead to an initial habituation effect, but when stress continues to build beyond a certain threshold, distress may resurface [27]. Although speculative, this trajectory is noteworthy because it has not been previously described in the context of the COVID-19 pandemic and may indicate an especially vulnerable group of healthcare workers who initially adapted well but experienced a resurgence of distress as the pandemic continued. However, this group comprised only 2% of the study population. While it was statistically supported in the expanded sample (two or more time points), it was no longer identified when restricting the analysis to participants with at least three time points. This may indicate that the smaller sample size constrained the model’s ability to detect the trajectory, rather than it being an artifact. Descriptively, individual cases following this pattern could still be observed in the dataset, warranting further investigation.

Our findings are consistent with several studies identifying trajectories of mental health in healthcare workers and the general population during the pandemic. As in most studies, the ‘resilience’ trajectory was the most pronounced [16]. It is noteworthy that the ‘resilience’ group still had a relatively high symptom burden compared to the general population.

Trajectories did not differ based on sociodemographic factors in our sample. Neither gender nor age predicted class membership. There were fewer physicians and more medical technical assistants in the ‘recovery’ group. Though non-significantly, medical technical assistants also had higher proportions of ‘delayed’, and lower proportions of ‘resilience’. Medical technical assistants, in our previous cross-sectional analyses, turned out to be the most distressed health profession at the beginning of the pandemic in Germany [28,29], possibly reflecting a sudden surge and urgency in laboratory analysis orders. Their form of being resilient, therefore, may rather have been expressed by a ‘recovery’ pattern than by a constantly low distress pattern.

Our data also show that the identified trajectories align well with self-reported retrospective assignments to typical trajectories. Most of the classifications matched well with the data. However, we also found that one-third of the ‘recovery’ group described themselves as still more burdened than before the pandemic. This reflects typical differences between external assessments and self-reports [30], but also indicates that the subjective experience of distress may not be sufficiently captured by depression and anxiety symptoms.

We labeled the trajectories of distress that we found in the data according to prior studies as ‘resilience’, ‘recovery’, and ‘delayed’ [15,31]. However, the current discussion on the term of resilience shows that a consensus on its definition as well as on the best means for its operationalization is still missing [32]. If resilience is defined as the ability to bend but not break in the face of adversity, to bounce back and sometimes even to grow stronger [33], then a trajectory of recovery could also be regarded as resilient, depending on the duration and level of distress. From a clinical or sociological point of view, it may even be healthy and favorable both for a learning effect and for possible change if a person reacts initially with distress to a distressing event. Also, there is no defined limit as to when a delayed distress response or recovery can still be regarded as related to a stressor. We therefore advise using the label ‘resilience’ with caution.

A further consideration concerns the predictive value of resilience scales. Most resilience scales are based on a trait concept [34], assuming that a predominantly stable pattern of cognitive, emotional, and social skills predicts a less distressed reaction when confronted with a stressor. Scales like the well-validated Resilience Scale ([21], short form RS-5 [35]) consequently show negative correlations with mental distress such as anxiety and depression. In our data, we could confirm this correlation. However, the resilience scale values were not stable over time, and could not predict the assignment to one of the typical distress trajectories. This result raises the question as to whether a resilience scale is of more predictive use for a more or less resilient stress trajectory than a simple distress measure. More research is needed on this topic, especially with comparisons of scales measuring resilience as a trait, a process, or an outcome [36].

Despite the comprehensive nature of this study, there are several limitations. Firstly, the absence of direct comparison data from before the pandemic limits our ability to fully understand the baseline stress levels and accurately classify the trajectories. This limitation is inherent to the study design and shared by comparable studies during the pandemic. Large cohorts or panels established over long timespans may be needed to assess the impact of non-predictable exceptional events. Secondly, while our sample size is relatively large for a longitudinal trajectory-based study, the extended duration and recruitment through open surveys led to expected fluctuations in participant retention over time. This means that despite the very large cross-sectional samples [29,37], the sample size decreased for longitudinal analyses. Additionally, as this is a self-selected sample, there may be biases that affect the generalizability of the findings. These limitations should be taken into account when interpreting the results and planning future research.

Our findings underscore the necessity of longitudinal approaches to understand the complex interplay of stressors and coping mechanisms during prolonged crises. Understanding these trajectories is crucial for informing targeted interventions to support the mental health of healthcare workers during the COVID-19 pandemic and future public health crises.

## 5. Conclusions

Our study provides valuable insights into the trajectories of mental distress and resilience among healthcare workers during the COVID-19 pandemic. By identifying three distinct patterns—resilience, recovery, and delayed distress—we highlight the heterogeneous nature of psychological responses to prolonged crises. While the majority of participants maintained low distress levels, a significant subset experienced either delayed or gradually improving symptoms, underscoring the need for tailored interventions. Importantly, our findings also challenge the predictive validity of brief resilience scales, emphasizing the complexity of resilience as a dynamic process rather than a stable trait. Future research should explore more nuanced measures of resilience and distress, while healthcare systems must prioritize long-term mental health support for frontline workers. By recognizing these trajectories, we can enhance preparedness for future public health crises and develop more effective, evidence-based psychological interventions.

## Figures and Tables

**Figure 1 healthcare-13-00574-f001:**
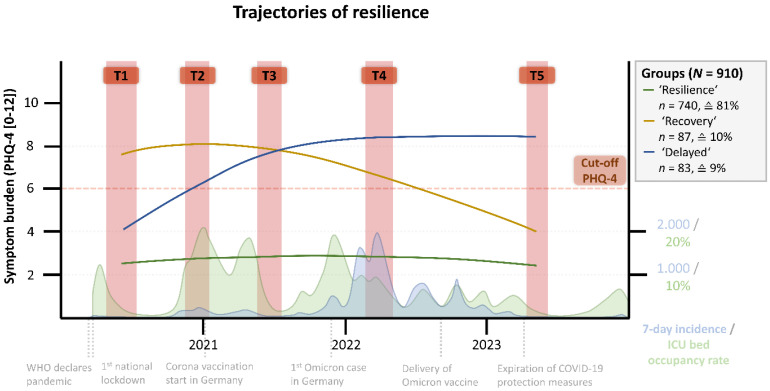
Trajectories of resilience among healthcare workers. All data and information pertain to Germany. Incidence rate and ICU bed occupancy rate are derived from the publicly available database of the Robert Koch Institute (www.rki.de). The cut-off value for the PHQ-4 is marked at 6 in the graph, with values above indicating major depressive and generalized anxiety disorder.

**Figure 2 healthcare-13-00574-f002:**
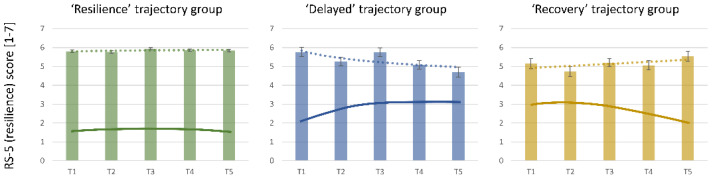
Trajectories of resilience based on symptom burden (PHQ-4) and resilience (RS-5). Scores of the RS-5 from T1 to T5 for the three empirical trajectories of resilience. The dashed line shows the progression over the various measurement times. The solid line shows the respective empirical trajectory, measured using the PHQ-4, for clarification.

**Figure 3 healthcare-13-00574-f003:**
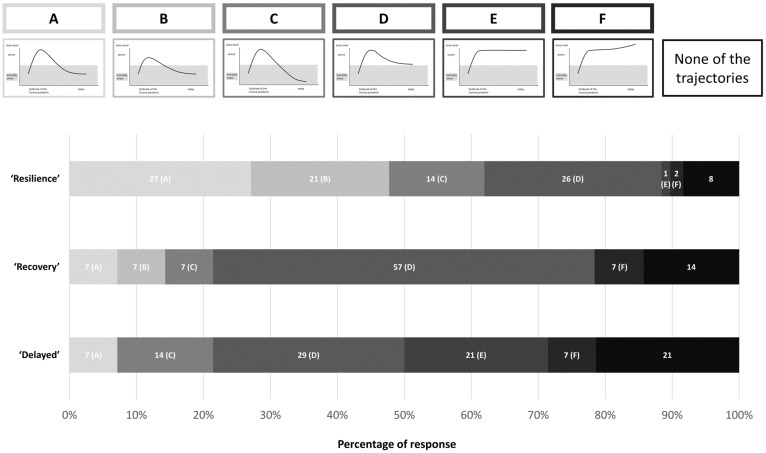
Self-reported trajectories of stress at T5 across the three data-based resilience trajectory groups. Panels (**A**–**F**) depict the predefined stress trajectories that participants could select to best represent their experience. The *x*-axis represents the time course of the pandemic, while the *y*-axis indicates self-reported stress levels. The rightmost box indicates the proportion of participants who felt that none of the provided trajectories accurately reflected their experience. The bar chart below shows the percentage of participants within each data-based trajectory (‘Resilience,’ ‘Recovery,’ and ‘Delayed’) who selected each self-reported trajectory.

**Table 1 healthcare-13-00574-t001:** Cohort composition (*n* = 910).

	Total	Resilience	Recovery	Delayed
Gender, *n* (%)				
Female	701	569 (81.17%)	71 (10.13%)	61 (8.70%)
Male	224	184 (82.14%)	17 (7.59%)	23 (10.27%)
Diverse	5	2 (40.00%)	1 (20.00%)	2 (40.00%)
Age, *n* (%)				
18–30	170	125 (73.53%)	23 (13.53%)	22 (12.94%)
31–40	268	208 (77.61%)	31 (11.57%)	28 (10.45%)
41–50	256	211 (82.42%)	19 (7.42%)	26 (10.16%)
51–60	325	280 (86.15%)	23 (7.08%)	22 (6.77%)
61–70	69	62 (89.86%)	3 (4.35%)	4 (5.80%)
70+	2	2 (100.00%)	0 (0.00%)	0 (0.00%)
Profession, *n* (%)				
Physicians	168	148 (88.10%)	8 (4.76%) *	12 (7.14%)
Nurses	231	184 (79.65%)	25 (10.82%)	21 (9.09%)
Medical Technical Assistants (MTAs)	156	112 (71.79%)	25 (16.03%) ***	18 (11.54%)
Psychologists	50	46 (92.00%)	3 (6.00%)	2 (4.00%)
Pastoral care workers	58	56 (96.55%)	1 (1.72%)	1 (1.72%)
Others	407	323 (79.36%)	40 (9.83%)	44 (10.81%)

The trajectory groups presented in this table were assigned based on model-derived classifications using Growth Mixture Modeling (GMM). There were fewer physicians (*p* = 0.048) * and more MTAs (*p* < 0.001) *** in the ‘recovery’ group than expected based on the overall sample distribution.

## Data Availability

The data supporting the findings of this study are available upon reasonable request from the corresponding author, subject to compliance with institutional data-sharing policies and applicable privacy or ethical restrictions.

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
