# Peer review of "Trajectories of Mental Distress and Resilience During the COVID-19 Pandemic Among Healthcare Workers"

_healthcare, 2025, doi:10.3390/healthcare13050574_

Round 1
Reviewer 1 Report
Comments and Suggestions for Authors
Overall excellent and interesting work! below are are my comments for each section.
Abstract
· Overall well written and does a nice job summarizing the key findings from the manuscript.
· The authors should it make it more clear if their GMM models adjusted for any additional factors such as age, gender, etc.
· The implication/context of being in each group is not clear
Introduction
· The introduction is well written and provides the needed context for the study. The authors also do a nice of clearly stating the aim of their research.
Methods
· Inclusion criteria is clearly stated, however the authors should make the exclusion criteria more explicit in the methods section.
· The authors make it clear who the link was available to, however it is not clear the total number of surveys administered, the response rates at each time point, or the loss to follow up over the five time periods.
· Table 1 and discussion around who ultimately were included (Section 2.2) in the study is better suited for the start of the results section.
· Also for Table 1, the authors should include in either the title or footnote whether the trajectory groups were derived from model assignment or self-reported.
· Statements for the results of the Cronbach’s alpha test should be included in the results section, not the methods section.
· Variables included in the models should be made clear
· On line 178, it is unclear for which factors internal consistency was measured.
Rasuls
· A paragraph is needed to describe the characteristics of the final sample.
· Unclear what the sentence from line 289 to line 291 is trying to state.
· The axis text on the stress graphs in Figure 3 is near impossible to read. The authors should try to reorganize these graphs to help that text be more legible or provide a description of the x axis and y axis in the footnote for this figure.
Discussion
· It is not fully clear or understandable to me when expanding the sample to individuals that only completed 2 measurements you find a new result that takes three measurements to assess (the ‘U’ shape as discussed from line 318 to 331). This discussion and result may need to be made clearer.
Reviewer 2 Report
Comments and Suggestions for Authors
- The manuscript is clear, relevant for the field and presented in a well-structured manner. The cited references are mostly recent publications (within the last 5 years) and appropriate. The manuscript’s results are reproducible based on the details given in the methods section. The conclusions are consistent with the evidence and arguments presented
- Small group of healthcare workers (N = 910) Line 124 vs. large sample size? Line 380
- “Others” group of the healthcare workers (N = 407) is big, subgroups are necessary for the analysis of this large group (46%) Line 133
- Figure 3 needs a make-over for the top items A to F: 6 figures on top are too small for interpretation Line 293
- Loss to follow-up: missing data for the 5 different points over time Line 382
- Therefore, use of participants who attended at least 2/5 and 3/5 time points: analysis with 4/5 time points maybe gives other results?
Reviewer 3 Report
Comments and Suggestions for Authors
Congratulations to the authors on their research. The choice of a growth mixture model methodology, the longitudinal design with five measurement points, and the long-term follow-up of healthcare professionals regarding distress and resilience are of great interest.
The following suggestions are proposed to improve the manuscript:
- Introduction: It would be beneficial to emphasize more strongly the need for long-term follow-ups of populations highly exposed to stress or trauma, such as healthcare workers in public health emergencies.
- Sample Characteristics: The manuscript should include the sample sizes for each measurement point, as well as the attrition rate or decrease in sample size across different time points.
- Sample Characteristics Section: The table presenting the cohort composition based on latent class analysis is somewhat confusing in this section. It is recommended that the authors keep the description of the main characteristics of the sample here and refer to the table in section 3.2. Cohort Characteristics.
- Statistical Analysis: The manuscript states that an LGMM analysis was conducted in R and that goodness-of-fit indices were evaluated. However, the parameters that guided the selection of a three-class model are not provided. It is suggested that the authors include this information. If space constraints are an issue, consider adding it as supplementary information.
- Results (Section 3.3): The manuscript states that there are differences between the resilience and recovery groups. However, in the following sentences, only differences between the recovery and delayed groups are presented. It would be useful to clarify whether these were post hoc analyses.
- U-Shaped Pattern: Both the Results and Discussion sections describe a U-shaped trajectory based on three measurement points. While this pattern represents only 2% of participants, making it already a limitation, it would be interesting to compare these findings with studies suggesting that cumulative exposure to stressful events may lead to a habituation effect, reducing psychological impact up to a certain threshold. That is, when stress continues accumulating—such as during successive waves of the pandemic—distress may resurface.
- For example, Seery, Holman, & Silver (2010) found quadratic relationships between cumulative adversity and psychological distress, depending on the number of adverse events experienced (Seery, M. D., Holman, E. A., & Silver, R. C. (2010). Whatever does not kill us: Cumulative lifetime adversity, vulnerability, and resilience. Journal of Personality and Social Psychology, 99(6), 1025–1041. https://doi.org/10.1037/a0021344)
- Discussion - Recovery vs. Resilience Debate: The manuscript raises an important debate regarding recovery and resilience trajectories, questioning whether recovery might, in fact, be another form of resilience. While the study is innovative in terms of its design, this debate is of great relevance. It is suggested that the authors further develop this discussion by comparing their findings with prior studies on responses to highly stressful or traumatic events (e.g., Bonanno, Hobfoll, or Galea).
Once again, congratulations on this valuable research, I found it very interesting.
